# Chilling Requirements of Apricot (*Prunus armeniaca* L.) Cultivars Using Male Meiosis as a Dormancy Biomarker

**DOI:** 10.3390/plants12173025

**Published:** 2023-08-23

**Authors:** Erica Fadón, Sara Herrera, Tudor I. Gheban, Javier Rodrigo

**Affiliations:** 1Departamento de Ciencia Vegetal, Centro de Investigación y Tecnología Agroalimentaria de Aragón (CITA), Gobierno de Aragón, Avda. Montañana 930, 50059 Zaragoza, Spaintigheban@cita-aragon.es (T.I.G.); 2Instituto Agroalimentario de Aragón—IA2, CITA-Universidad de Zaragoza, Calle Miguel Servet 177, 50013 Zaragoza, Spain

**Keywords:** Chilling Hours, Chilling Units, Chill Portions, flower primordia, fluorescence microscopy

## Abstract

Apricot has undergone an important cultivar renewal during the last years in response to productive and commercial changes in the crop. The impact of the sharka disease (plum pox virus) prompted the release of cultivars resistant/tolerant to this virus, leading to a major cultivar renewal worldwide. This has caused high variability in chilling requirements on new releases that remain unknown in many cases. In many apricot-growing areas, the lack of winter chilling is becoming a limiting factor in recent years. To deal with this situation, growers must choose cultivars well adapted to their areas. However, the information available on the agroclimatic requirements of the cultivars is very limited. To fill this gap, in this work, we have characterized the chilling requirements of 13 new apricot cultivars from Europe (France, Greece and Spain) and North America (USA) in two experimental collections in Aragón (Spain). We established the chilling period using male meiosis as a biomarker for endodormancy release over two years. Chilling requirements ranged from 51.9 Chill Portions (CP) to 70.9 CP. Knowing the chilling requirements of cultivars will help growers to select suitable cultivars adapted to the chill availability of their region.

## 1. Introduction

Apricot, as other temperate fruit tree species, presents cultivar-specific chilling requirements to overcome winter dormancy [1]. Dormancy allows the adaptation of these species to temperate climates, since this allows them to survive the unfavorable conditions of winter [2,3]. During dormancy, the meristems neither show growth nor development, and remain protected inside the buds, being resistant to cold and even to freezing temperatures [1]. In the first phase, growth is arrested by internal factors and the exposition to chill during a certain period is needed to restore growth capacity (endodormancy). In the second phase, growth is arrested because of the unsuitable weather conditions, and the buds require the exposition to warm temperatures to finally resume bud growth (ecodormancy) [4]. The chilling requirements delineate the cultivation area of each cultivar, but despite its importance, this parameter is unknown for most of the apricot varieties currently cultivated [5].

Traditionally, to differentiate endo- from ecodormancy both experimental [6] and statistical methodologies [7,8] have been used. The experimental methodology, or forcing test, consists of exposing shoots in a growth chamber sequentially during winter, thus to different chilling exposures. Then, bud growth is evaluated after a certain period in the warm conditions [6]. The statistical approach estimates the date of chilling fulfillment correlating long series of phenological observations with the temperature records of the previous season [7,8]. These methods are extensively applied but present important limitations that limit the characterization of the agroclimatic requirements of the cultivars. On one side, the forcing test is cumbersome and time-consuming [9,10], and the lack of standardization in the experimental designs often leads to inconsistencies in the data obtained [3,5]. On the other hand, the statistical methodology requires more than 20 years of data, which often is difficult to obtain [7,8].

The lack of knowledge on the biological process behind dormancy has prevented an accurate delineation of dormancy phases [11]. Several efforts have been made to find a biomarker for endodormancy [1]. In temperate fruit trees, flowers develop along the seasons, with a specific rhythm for each species. Flower induction and early flower differentiation occur during the summer and autumn, then growth is arrested during the winter [12,13]. Then, no morphological changes are shown until growth resumption in late winter or early spring that continue until flowering [14,15,16,17].

In temperate *Prunus* sp., male meiosis occurs synchronized with the seasons and relates with dormancy phases [18,19]. At dormancy establishment of flower buds during the autumn, the anthers show the sporogenous tissue composed of the pollen mother cells, surrounded by the tapetum and the middle layers. The first changes to be detected after endodormancy breaking are related with male meiosis, i.e., a callose layer is formed surrounding the pollen mother cells until they are completely isolated from each other. Then, they undergo the two meiosis divisions giving rise to four haploid cells separated by callose inner walls, forming the characteristic tetrads. Later, the tapetum produces an enzyme that degrades the callose (callase) releasing the microspores that continue their development to mature pollen grains [17,20]. Although the phases of pollen development are conserved among the species, the timing in relation to dormancy varies [18,19]. In apricot, the onset of male meiosis has been associated with the transition from endo- to ecodormancy [16,17], since it establishes a limit between the sporogenous tissue and the initial stages of formation of pollen grains [18,21]. In recent years, male meiosis has shown to be an adequate biomarker to determine the end of endodormancy in apricot [18,22], facilitating the delineation of dormancy and the estimation of both chilling and forcing requirements [23]. In apricot, the sporogenous tissue is presented during endodormancy, and pollen development is resumed during ecodormancy [24], where meiosis indicates the transition among the endo- and ecodormancy phases [17,18,23].

Apricot world production has reached 3.7 million tons, and it is considered one of the most economically important fruit crops in temperate regions [25]. In recent years, an important renewal of apricot cultivars is occurring worldwide [26] because the sharka disease, caused by the plum pox virus (PPV), has limited the cultivation of apricot in many areas [27]. Many breeding programs have incorporated PPV-resistance using apricot cultivars resistant/tolerant to this virus from North America as parents, which also have high chilling requirements [28]. As a result, the new releases have a wider range of chilling requirements than the well-adapted traditional cultivars. The lack of characterization of the chilling requirements of these new releases is limiting the optimal selection of cultivars for each growing area [1]. In this work, we aim to characterize the chilling requirements of 13 cultivars of apricot using male meiosis as a biomarker for endodormancy release of flower buds.

## 2. Results

### 2.1. Temperature Regimes and Chill Accumulation

The temperature regimes during the dormancy months (from October to March) were very similar in both locations (Figure 1A,B). The temperature threshold was wider in Caspe, with higher temperatures during the autumn (maximum temperatures about 18 °C) and lower temperatures during winter (minimum temperatures reaching 5 °C in January) (Figure 1A) and between 15 °C and 7 °C in La Almunia de Doña Godina (Figure 1B). The year 2018 was slightly colder than 2019, and this resulted in a difference of chill accumulation of about eight Chill Portions (CP) in February for the same date in both years (Figure 1C,D).

### 2.2. Endo- to Ecodormancy Transition

We identified four main stages of the male meiosis through the microscopy observations of the anthers in 13 apricot cultivars released from eight breeding programs of Europe and North America (Table 1).

In the first stage, the sporogenous tissue, composed of pollen mother cells, was observed in the earlier sampling dates that were considered the endodormancy phase (Figure 2A). Then, each pollen mother appeared isolated by a surrounding callose layer; thus, meiosis was ongoing (Figure 2B). The process of meiosis terminated with the division into four meiotic subproducts with the formation of callose inner walls (tetrads) (Figure 2C). The presence of callose as an indicator of male meiosis occurring was considered the transition from endo- to ecodormancy. Finally, the young microspores were released after the degradation of the callose layers at ecodormancy (Figure 2D). This process lasted up to about 5 days in the different apricot cultivars analyzed.

In Caspe, the dates of male meiosis ranged between 8 January and 14 February 2018 and between 23 January and 10 February 2019 (Figure 3). In La Almunia de Doña Godina, the dates ranged between 21 January and 30th for ‘Sunny Cot’ and ‘Fartoli’ in 2018 and between 23 January and 10 February for ‘Sweet Cot’ and ‘Farely’ in 2019 (Figure 4). In general, the dates of male meiosis were earlier in 2018 than in 2019, specially for the cultivars ‘Farely’ (Figure 4) and ‘Colorado’ (Figure 3), with an elapse of 16 and 15 days, respectively. On the other hand, ‘Fartoli’ showed the same dates for male meiosis in both years (Figure 4), and for ‘Orange Red’, it occurred 4 days earlier in 2019 (9 February) than in 2018 (12 February) (Figure 3).

### 2.3. Chilling Requirements of 13 Apricot Cultivars

The cultivars ‘Colorado’ (52 ± 12 CP, 769 ± 156 CH, 963 ± 269 CU) (Table 2) and ‘Sunny Cot’ (57 ± 6 CP, 783 ± 32 CH, 1016 ± 97 CU) (Table 3) presented the lower chilling requirements in Caspe and La Almunia, respectively. While the cultivars ‘Orange Red’ (72 ± 4 CP, 1101 ± 63 CH, 1402 ± 54 CU) (Table 2) and ‘Farely’ (64 ± 13 CP, 897 ± 83 CH, 1172 ± 246 CU) (Table 3) presented the higher chilling requirements in Caspe and La Almunia, respectively.

The chilling requirements ranged between 52 CP and 72 CP, so the cultivars can be grouped in low chilling requirements from 52 CP to 58 CP (‘Colorado’, ‘Magic Cot’, ‘Sunny Cot’, ‘Mirlo Blanco’, ‘Bigred’, ‘Goldbar’, ‘Sweet Cot’ and ‘Pinkcot’), medium chilling requirements with 63–64 CP (‘Fartoli’, ‘Delice Cot’ and ‘Farely’) and high chilling requirements from 67 CP to 72 (‘Hardgrand’ and ‘Orange Red’) (Table 2 and Table 3).

## 3. Discussion

In this work, we established the chilling requirements of 13 apricot cultivars, 12 of them for the first time, using male meiosis as a biomarker of overcoming endodormancy. This approach provides a more objective and direct methodology to establish the agroclimatic requirements of apricot cultivars than previous methodologies.

### 3.1. Male Meiosis and Endodormancy Breaking

Our observations of apricot anthers under the microscope allowed identifying of the key developmental stages of male meiosis, a conserved process in plants [29,30]. The emergence of callose around the microspore mother cells allowed the identification of the boundary between endo- and ecodormancy. Callose accumulation is a physical filter of molecules in response to degeneration processes, such as ovule abortion [31,32] or various types of stress [33]. Cold winter temperatures clearly influence pollen meiosis time, which in turn is reflected in flowering time [18].

The differences observed in the dates of male meiosis were mainly due to the diversity of the cultivars and the chill variation between years, since the two locations presented very similar temperature regimes during the dormancy months. These results fit with previous studies that characterized the male meiosis of apricot cultivars in Zaragoza (Spain), very close to the cultivar collections that we have worked on for this study. In a previous work, the meiosis dates for cultivars ‘Canino’, ‘Corbato’, ‘Moniquí’, ‘Paviot’ and ‘Luizet’ occurred in a similar period, between 25 January and 10 February [18]. A later study established the dates of male meiosis of 20 cultivars over 8 years in a similar range of dates, between 14 January and 8 February [23]. In another study in Hungary, eight apricot cultivars were ranked based on microsporogenesis dates over three years, founding the same order of meiosis time between cultivars in all years [21]. The influence of winter temperatures on both pollen meiosis and flowering time was also found when microsporogenesis was analyzed in three Hungarian apricot for 24 years, finding high differences in the meiosis dates between cultivars and years. The advances in both meiosis and flowering time observed over the years were related to the increasingly milder winter temperatures caused by regional warming of the climate in the Carpathian Basin [34].

### 3.2. Chilling Requirements

The chilling requirements ranged from ‘Colorado’ (52 ± 12 CP, 769 ± 156 CH, 963 ± 269 CU) to ‘Orange Red’ (72 ± 4 CP, 1101 ± 63 CH, 1402 ± 54 CU). These results are in line with previous studies in other apricot cultivars. Until now, 68 apricot cultivars have been characterized in 15 studies with the experimental forcing test, which resulted in ranges of chilling requirements of 30–79 CP, 171–1812 CH and 274–1665 CU [5]. A subsequent study characterized another 20 apricot cultivars both with the forcing test and two statistical methods, establishing the chilling requirements between 33.9 ± 4.6 CP and 76.7 ± 4.2 CP [23].

Several studies have estimated the date of endodormancy breaking and the chilling requirements of ‘Orange Red’ with the forcing test in several locations (Spain, Italy and South Africa) [35,36,37,38,39,40]. Our estimations of endodormancy breaking of the cultivar ‘Orange Red’ were 9 and 12 February that resulted in later dates than those observed in Murcia (Spain), between 21 January and 26 February [41], and in Piemonte (Italy), between 17 January–2 February [39,40]. Based on the date of endodormancy breaking, we estimated the chilling requirements of ‘Orange Red’ as 72 ± 4 CP, 1101 ± 63 CH and 1402 ± 54 CU. These results are in line with the studies performed in Murcia (Spain), especially when using the Dynamic Model to quantify chill (69.1 ± 3 CP) [38] or (64 ± 9.9 CP) [35]. Our results are also close to those from Italy quantifying chilling with the Utah Model (1450 CU) [36]. However, our results are extremely different from those obtained in South Africa (55.4 ± 4.5 CP, 568 ± 3.6 CH, and 957 ± 9.6 CU) or in Murcia using the Chill Hour Model (777 ± 12.8 CH) [35]. There are multiple factors that caused the wide dispersion of the data even when obtained with the forcing test. On one side, the models do not fit among the different climates, and on the other side, the protocols present variations among the different working groups [23,38,40]. On this point, the use of a dormancy biomarker presents the main advantage of being a biological process that can be objectively determined [5].

### 3.3. Male Meiosis as a Biomarker for Dormancy

Male meiosis in apricot is the first biomarker to determine dormancy phases and estimate chilling requirements [23]. We applied this methodology that has numerous advantages compared to the forcing test or the statistical protocols for dormancy delineation. The main advantage is that this method is based on a biological process, which is an objective parameter that is directly identified; thus, it does not depend on protocols with many variables as is the case with the forcing test, whose results can be altered depending on the conditions of the forcing chamber or the criteria used for the evaluation of bud growth [5,6]. This method also has advantages over the statistical methodology, which depends on the interpretation of the results and the criteria for establishing the limits of the parameters [7,8].

In this study, we based our work on fixed material previously sampled that were analyzed at a later time. However, the developmental stage of the anthers could be directly determined in fresh material. This alternative would allow obtaining the results of chilling requirements or chilling fulfilment in a very short time, which would allow making decisions in the management of the orchard even in the same season. Neither the empirical nor the statistical methods could provide the information in such a very short time; furthermore, the statistical methodology requires more than 20 years of data [5,7,8] that are very difficult to obtain for most cultivars, especially for recent releases. This fact, along with additional advantages of the protocol, such as the very few flower buds needed or the easy microscopy process for a trained analyst, make this approach suitable for being incorporated in breeding programs to directly characterize the chilling requirements or even for orchard management.

Male meiosis has been demonstrated to be an effective biomarker for dormancy in apricot based on previous studies on anther development and dormancy [17,18]. However, studies in sweet cherry indicated that male meiosis occurs several weeks after dormancy release, and therefore, it is not an adequate dormancy biomarker in that species [19,20]. Further research is needed to explore if the close relationship between male meiosis and endodormancy breaking in apricot is also conserved in other temperate fruit tree species.

## 4. Materials and Methods

### 4.1. Plant Material

Thirteen cultivars of apricot were selected from two cultivar collections located in Caspe, Zaragoza (‘Bigred’, ‘Colorado’, ‘Hargrand’, ‘Magic Cot’, ‘Mirlo Blanco’, ‘Orange Red’ and ‘Pinkcot’) and in La Almunia de Doña Godina, Zaragoza (‘Delice Cot’, ‘Farely’, ‘Fartoli’, ‘Goldbar’, ‘Sunny Cot’ and ‘Sweet Cot’).

Ten flower buds were sampled weekly from December to February over 2 years (2018–2019). The buds were fixed in ethanol (95%)/acetic acid (3:1, *v*/*v*) for 24 h and conserved in ethanol 75% at 4 °C [42].

### 4.2. Male Meiosis as a Biomarker for Determination of Dormancy Breaking 

The dates of the male meiosis for each cultivar and year were determined through fluorescence microscopical observations. First, we removed the anthers from the fixed buds with the help of a scalpel under a stereoscopic microscope and a clock glass with ethanol at 75%. Then, we placed the anthers over a slide, added a drop of 0.1% aniline blue in 0.1 NK_3_PO_4_ for callose staining and squashed the anther to facilitate the contact of the stain with the cells inside the locules of the anthers [23]. These preparations were directly observed under a Leica DM2500 microscope (Cambridge, UK) with UV epifluorescence using a 340–380 bandpass and 425 longpass filters. According to Julian et al. (2014) [18], the formation of a callose wall around the microspore mother cells indicated the occurrence of meiosis.

### 4.3. Chill Quantification

Temperature data were obtained from the “Oficina del Regante”, a platform that manages the temperature records of the meteorological stations in Aragón, Spain [43]. We downloaded the hourly temperatures of the meteorological stations at Caspe and at Epila for La Almunia de Doña Godina.

The chill accumulation was computed during the chilling period according to the three models most widely used in dormancy-related studies, as well as in temperate orchard management. The Chilling Hours Model defined a “Chilling Hour” (CH) as one hour within a temperature range between 0 and 7.2 °C [44]. The Utah Model computed “Chilling Units” (CU) using different chill effectiveness weights corresponding to various temperature ranges [45]. Finally, the Dynamic Model accumulates “Chill Portions” (CP) through a two-step process in which a chilling precursor is formed in cool conditions and later converted to a permanent CP through a subsequent process that shows optimal effectiveness at mild temperatures [46,47,48]. The calculations were conducted using chillR v.0.72.8 package [49] in the R v.4.3.0 programming environment [50].

## 5. Conclusions

Male meiosis has been demonstrated to be an effective biomarker for dormancy in apricot based on previous studies on anther development and dormancy [17,18,23]. Subsequently, the observation of male meiosis under the microscope was validated to establish endodormancy breaking dates in apricot cultivars, emerging as a better alternative than the other available methods, both experimental and statistical [23]. It has the additional advantage that flower buds can be analyzed immediately after collection, avoiding the variability of results caused by the multiple factors that can affect forcing experiments [9,51] and without the need to have a large set of phenological data for statistical analyses [7,11]. However, further work is needed to explore whether the close relationship between male meiosis and endodormancy breaking in apricot is also conserved in other temperate fruit tree species since very little information is available. In sweet cherry, male meiosis occurs several weeks after endodormancy release and therefore cannot be used as a suitable dormancy biomarker in that species [19,20]. The close relationship between male meiosis and endodormancy breaking found in apricot can probably also be conserved in other temperate fruit tree species with a shorter forcing phase than sweet cherry. In addition, the use of male meiosis as a biomarker for the delineation of dormancy phases can also be applied for physiological and genetic studies of dormancy as well as to determine if flower buds are vulnerable to frosts, since at the beginning of ecodormancy flower buds fail to show appreciable external signs of development but are increasingly vulnerable to low temperatures once their chilling requirements have been fulfilled [52,53,54,55].

## Figures and Tables

**Figure 1 plants-12-03025-f001:**
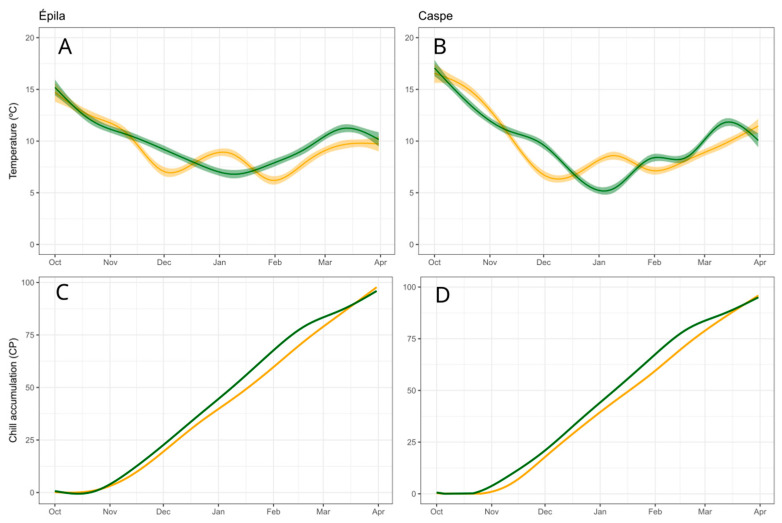
Temperature in (**A**) La Almunia de Doña Godina and in (**B**) Caspe. Chill accumulation in (**C**) La Almunia de Doña Godina and in (**D**) Caspe from October to April in the seasons 2017–2018 (green) and 2018–2019 (yellow).

**Figure 2 plants-12-03025-f002:**
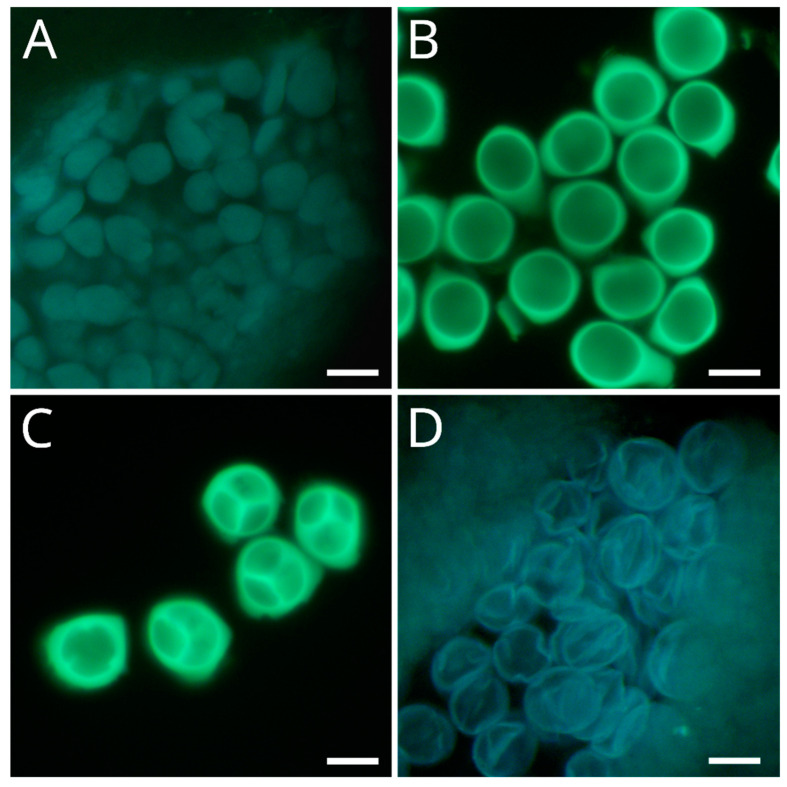
Microscope observations of apricot anthers. (**A**) Sporogenous tissue composed of the pollen mother cells. (**B**) Pollen mother cells surrounded by callose while meiosis was ongoing. (**C**) Tetrads and inner callose walls separate the four meiotic products. (**D**) Young microspores released.

**Figure 3 plants-12-03025-f003:**
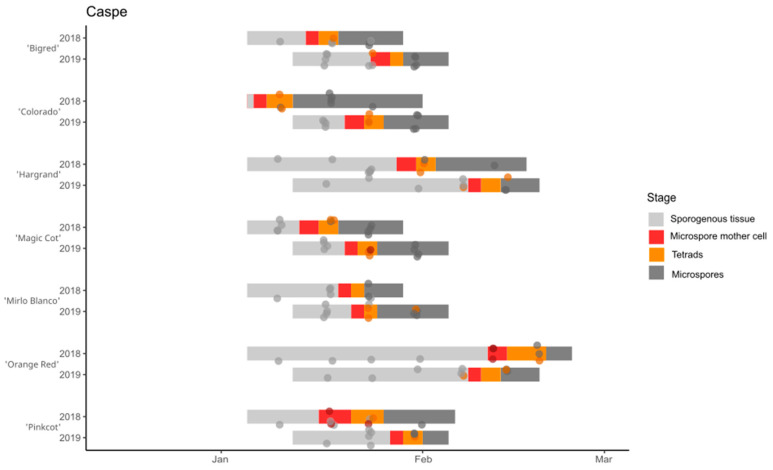
The anther developmental stages for the seven apricot cultivars grown in Caspe for two years (2018 and 2019). The dots indicate the observations, and the bars indicate the estimation of each developmental stage.

**Figure 4 plants-12-03025-f004:**
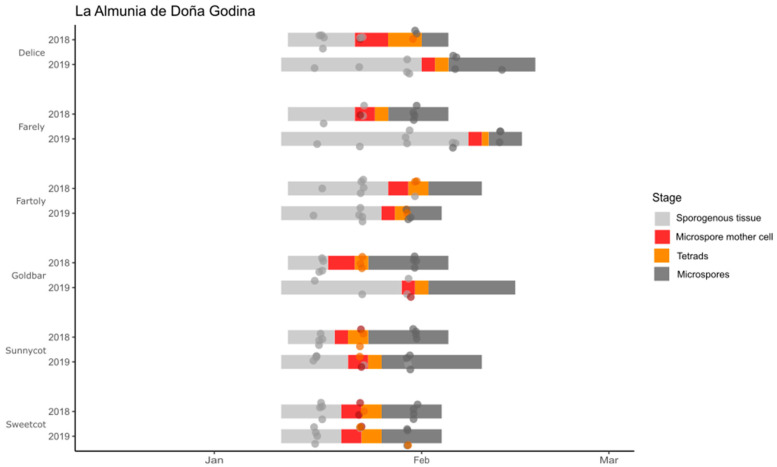
The anther developmental stages for the six apricot cultivars grown in La Almunia de Doña Godina during two seasons (2018–2019). The dots indicate the observations, and the bars indicate the estimation of each developmental stage.

**Table 1 plants-12-03025-t001:** Origin of the 13 cultivars analyzed in this study.

Breeding Program	Cultivar
Benoît Escande Edition BEE (France)	‘Bigred’ ‘Pinkcot’
CEBASfruit (Spain)	‘Mirlo Blanco’
Marie-France Bois, International Plant Selection (IPS)	‘Farely’ ‘Fartoli’
Marie-Laure Eteve (France), Cot International	‘Delice Cot’
PSB Producción Vegetal (Spain)	‘Colorado’
Rutgers (USA)	‘Orange Red’
SDR Fruit LLC (USA) New Jersey Agricultural Experiment Station	‘Magic Cot’ ‘Sunny Cot’
Washington State University Research Foundation (USA), Cot International	‘Goldbar’ ‘Sweet Cot’

**Table 2 plants-12-03025-t002:** Dates of male meiosis, indicating the overcoming of endodormancy, and chilling requirements of seven apricot cultivars grown in Caspe.

Caspe				
Cultivar	Meiosis Date	Chilling Portions	Chilling Hours	Chilling Units
‘Bigred’	16 January 2018	57 ± 10	846 ± 98	1065 ± 216
	27 January 2019			
‘Colorado’	8 January 2018	52 ± 12	769 ± 156	963 ± 269
	23 January 2019			
‘Hargrand’	31 January 2018	67 ± 11	988 ± 96	1275 ± 234
	10 February 2019			
‘Magic Cot’	16 January 2018	55 ± 8	824 ± 67	1024 ± 159
	22 January 2019			
‘Mirlo Blanco’	21 January 2018	57 ± 6	839 ± 57	1055 ± 139
	23 January 2019			
‘Orange Red’	14 February 2018	72 ± 4	1101 ± 63	1402 ± 54
	10 February 2019			
‘Pinkcot’	21 January 2018	59 ± 9	865 ± 93	1107 ± 213
	29 January 2019			

**Table 3 plants-12-03025-t003:** Dates of male meiosis, indicating the overcoming of endodormancy, and chilling requirements of seven apricot cultivars grown in La Almunia de Doña Godina.

La Almunia de Doña Godina				
Cultivar	Meiosis Date	Chilling Portions	Chilling Hours	Chilling Units
‘Delice Cot’	27 January 2018	63 ± 10	881 ± 15	1159 ± 164
	3 February 2019			
‘Farely’	25 January 2018	64 ± 13	897 ± 83	1172 ± 246
	10 February 2019			
‘Fartol’	30 January 2018	63 ± 6	873 ± 49	1144 ± 93
	30 January 2019			
‘Goldbar’	22 January 2018	58 ± 6	795 ± 16	1034 ± 106
	31 January 2019			
‘Sunny Cot’	21 January 2018	57 ± 6	783 ± 32	1016 ± 97
	24 January 2019			
‘Sweet Cot’	23 January 2018	59 ± 8	803 ± 8	1052 ± 119
	23 January 2019			

## Data Availability

Not applicable.

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
