# Peer review of "Chilling Requirements of Apricot (Prunus armeniaca L.) Cultivars Using Male Meiosis as a Dormancy Biomarker"

_plants, 2023, doi:10.3390/plants12173025_

Round 1
Reviewer 1 Report
The manuscript adds a significant knowledge to understanding and detecting chilling requirements of apricot cultivars.
Some suggestions:
1) Introduction. May be emphasized that this method is suitable for generative bud endodormancy detection, not the whole plant or vegetative buds. Moreover, scientific experience about use of male meiosis as the biomarker should be given. I advice move the paragraph 3.1. from Discussion to Introduction.
2) Results. Line 101 - should be "Figure 2B". Lines 140 - 141 - delete extra text.
3) Discussion. Move the 1st paragraph to Introduction.
4) Conclusions. Please add the main outcome of this particular study.
Author Response
Response to Reviewer 1 Comments
Remarks: The manuscript adds a significant knowledge to understanding and detecting chilling requirements of apricot cultivars.
Response: We do appreciate the comprehensive review that has clearly contributed to improve our paper over our original submission. The paper has been revised giving careful consideration to the points raised by the reviewer. We describe in detail below the changes made in each point.
Point 1: Introduction. May be emphasized that this method is suitable for generative bud endodormancy detection, not the whole plant or vegetative buds. Moreover, scientific experience about use of male meiosis as the biomarker should be given. I advice move the paragraph 3.1. from Discussion to Introduction.
Response 1:
- The fact that this method is suitable for flower buds is emphasized in the last sentence of ‘Introduction’ as suggested (lines 89-90)
- Several references have been given showing the scientific experience on the use of male meiosis as a biomarker (new references 1, 3, 15, 16, 21, 22, 34).
- The paragraph has been moved from Discussion to Introduction (lines 60-70).
Point 2: Results. Line 101 - should be "Figure 2B". Lines 140 - 141 - delete extra text.
Response 2:
- “Figure 2D” has been corrected to “Figure 2B” (line 114).
- Extra text has been deleted in the Table 2 caption (line 154).
Point 3: Discussion. Move the 1st paragraph to Introduction.
Response 3:
- As mentioned above, the paragraph has been moved from Discussion to Introduction (lines 60-70).
Point 4: Conclusions. Please add the main outcome of this particular study.
Response 4:
- The ‘Conclusions’ section has been revised and expanded, including the main outcome of this study.

Reviewer 2 Report
Apricot is an important temperate fruit species. In order to grow successfully, we need to know the physiological processes of the different genotypes, their environmental requirements, and their responses to changes in environmental factors. The changed environmental conditions can result in effects that are of great economic importance. In the cultivation of apricot, frost damage in winter and spring mean significant risks. Frost tolerance is closely related to the rate of development of overwintering organs, the length of endodormancy, and the inherited chilling and heat requirements of the cultivar. Many research results have been produced in this area, but every new result is important.
The new results presented in the article provide a lot of new, valuable information in this research area.
However, due to mistakes, the article is only suitable for publication after correction.
Mistakes for example:
There are typos in the text, e.g. in line 285. You have to check the text accurately from this aspect.
Reference 14 is in wrong place in the text.
In table 3, there are unnecessary instructions in the text.
The authors are not up-to-date in their knowledge of the literature, many important references are missing.
The following papers must be referenced in the article:
Andreini, L., Viti, R., Bartolini, S, Ruiz, D., Egea, J., Campoy, J.A. 2012. The relationship between xilem differentiation and dormancy evolution in apricot flower buds (Prunus armeniaca L.): the influence of environmental conditions in two Mediterranean areas. Trees. 26:919-928.
Andreini, L., Cortazar-Atauri, I.G., Chuine, I., Viti, R., Bartolini, S., Ruiz, D., Campoy, J.A., Legave, J.M., Audergon, J.M., Bertuzzi, P. 2014. Understanding dormancy release in apricot flower buds (Prunus armeniaca L.) using several process-based phenological models. Agricultural and Forest Meteorology. 184:210-219.
Bartolini, S., Viti, R., Guerriero, R. 2006a. Xilem differentiation and microsporogenesis during dormancy of apricot flower buds (Prunus armeniaca L.) European J. of Hort. Sci. 71:84-90.
Bartolini, S., Viti, R., Laghezali, M., Olmez, H.A. 2006b. Xilem vessel differntiation and microsporogenesis evolution in ’Canino’ cultivar growing in tree different climatic areas: Italy, Marocco and Turkey. Acta Hort. 701:135-140.
Hajnal V., Omid, Z., Ladányi, M., Tóth, M., Szalay, L. 2013. Microsporogenesis of apricot cultivars in Hungary. Not Bot Horti Agrobo, 41(2):434-439.
Szalay, L., Froemel-Hajnal, V., Bakos, J., Ladányi, M. 2019. Changes of the microsporogenesis process and blooming time of three apricot genotypes (Prunus armeniaca L.) in Central Hungary based on long-term observation (1994-2018). Scientia Horticulturae. 246:279-288. https://doi.org/10.1016/j.scienta.2018.09.069.
Viti, R., Bartolini, S., Andreini, L. 2013. Apricot flower bud dormancy: Main morphological, antomical and physiological features related to winter climate influence. Adv. Hort. Sci. 27(1-2):5-17.
Author Response
Response to Reviewer 2 Comments
Remarks: Apricot is an important temperate fruit species. In order to grow successfully, we need to know the physiological processes of the different genotypes, their environmental requirements, and their responses to changes in environmental factors. The changed environmental conditions can result in effects that are of great economic importance. In the cultivation of apricot, frost damage in winter and spring mean significant risks. Frost tolerance is closely related to the rate of development of overwintering organs, the length of endodormancy, and the inherited chilling and heat requirements of the cultivar. Many research results have been produced in this area, but every new result is important.
The new results presented in the article provide a lot of new, valuable information in this research area.
However, due to mistakes, the article is only suitable for publication after correction.
Response: We do appreciate the comprehensive review that has clearly contributed to improve our paper over our original submission. The paper has been revised giving careful consideration to the points raised by the reviewer. We describe in detail below the changes made in each point.
Point 1:
There are typos in the text, e.g. in line 285. You have to check the text accurately from this aspect.
Response 1:
- The text has been carefully revised, correcting this (line 298) and other typos.
Point 2: Reference 14 is in wrong place in the text.
Response 2:
- All references have been carefully revised, both in the text and the list, includng a number of new citations. All of them have been revised and renumbered.
Point 3: In table 3, there are unnecessary instructions in the text.
Response 3:
- Extra text has been deleted in the Table 2 caption (line 154).
Point 4: The authors are not up-to-date in their knowledge of the literature, many important references are missing.
The following papers must be referenced in the article:
Andreini, L., Viti, R., Bartolini, S, Ruiz, D., Egea, J., Campoy, J.A. 2012. The relationship between xilem differentiation and dormancy evolution in apricot flower buds (Prunus armeniaca L.): the influence of environmental conditions in two Mediterranean areas. Trees. 26:919-928.
Andreini, L., Cortazar-Atauri, I.G., Chuine, I., Viti, R., Bartolini, S., Ruiz, D., Campoy, J.A., Legave, J.M., Audergon, J.M., Bertuzzi, P. 2014. Understanding dormancy release in apricot flower buds (Prunus armeniaca L.) using several process-based phenological models. Agricultural and Forest Meteorology. 184:210-219.
Bartolini, S., Viti, R., Guerriero, R. 2006a. Xilem differentiation and microsporogenesis during dormancy of apricot flower buds (Prunus armeniaca L.) European J. of Hort. Sci. 71:84-90.
Bartolini, S., Viti, R., Laghezali, M., Olmez, H.A. 2006b. Xilem vessel differntiation and microsporogenesis evolution in ’Canino’ cultivar growing in tree different climatic areas: Italy, Marocco and Turkey. Acta Hort. 701:135-140.
Hajnal V., Omid, Z., Ladányi, M., Tóth, M., Szalay, L. 2013. Microsporogenesis of apricot cultivars in Hungary. Not Bot Horti Agrobo, 41(2):434-439.
Szalay, L., Froemel-Hajnal, V., Bakos, J., Ladányi, M. 2019. Changes of the microsporogenesis process and blooming time of three apricot genotypes (Prunus armeniaca L.) in Central Hungary based on long-term observation (1994-2018). Scientia Horticulturae. 246:279-288. https://doi.org/10.1016/j.scienta.2018.09.069.
Viti, R., Bartolini, S., Andreini, L. 2013. Apricot flower bud dormancy: Main morphological, antomical and physiological features related to winter climate influence. Adv. Hort. Sci. 27(1-2):5-17.
Response 4: Many thanks for the suggested citations. All of them have now been included in the manuscript (references 1, 3, 15, 16, 21, 22, 34).

Reviewer 3 Report
Dear Fadón et al.,
I have read and reviewed your manuscript ‘chilling requirements of apricot cultivars using male meiosis as a dormancy marker’ with pleasure. The research provides an interesting insight in the determination of chilling requirement and offers useful tools for breeders in the field.
I have some comments on the order in which the results and discussion are written down and would add some suggestions for change. Other than that, all that’s left for me to add are some minor comments on spelling and syntax, that I will outline below.
Introduction
Line 31. ‘…temperate climates, since this allows them to survive the unfavourable conditions of winter.’
Line 32. ‘…the meristems neither show growth nor development…’
Line 51. ‘…on the other hand, the statistical…’
Line 53. ‘…process behind dormancy has prevented an accurate delineation…’
Line 54. ‘…efforts have been made…’
Line 55. ‘…dormancy release during flower development...’? I’m not entirely sure what the authors are trying to say in this sentence.
Results
Line 86. Though it is elaborated upon in the material and methods, for clarity I’d like to see a brief explanation of the ‘chilling portion’ when it is first mentioned.
Line 115. More principally, can one say that the ‘cultivars and seasons were the main causes of the variation’ when you don’t grow the specimens on both locations? I think one should grow all cultivars on both geographical locations to say whether location has an effect.
Line 140. There’s a sentence to be removed, from ‘this is a table…’ onwards.
Discussion
Line 177. ‘…completely isolated from each other…’
Line 180. ‘…that continue their development to mature pollen…’
Line 183. ‘…resumed during ecodormancy, where meiosis indicates the transition…’
Line 239. ‘…difficult to obtain…’ as alternative to ‘…difficult to be available…’
In general, please be consistent in the use of ‘chill’ versus ‘chilling’ requirements (ignore this if I’m wrong, but I have the impression that they both are used for the same definition). I think you use ‘chilling requirements’ more regularly but I’ve come across some ‘chill’ too.
The discussion starts by the statement that ‘this approach provides a more objective and direct methodology to establish the agroclimatic requirements of apricot cultivars than previous methodologies’. While I agree with the statement, I think that it would be helped by the results that are now presented in lines 203-219. Is there a way to present the comparison in the ‘Results’ section? Paragraph 203-219 presents new data, which I believe is well suited for the ‘Results’ and it is further confirmation that the experiments/methodology is sound.
Yours truly,
See the 'comments and suggestions for authors' section
Author Response
Response to Reviewer 3 Comments
Remarks:
I have read and reviewed your manuscript ‘chilling requirements of apricot cultivars using male meiosis as a dormancy marker’ with pleasure. The research provides an interesting insight in the determination of chilling requirement and offers useful tools for breeders in the field.
I have some comments on the order in which the results and discussion are written down and would add some suggestions for change. Other than that, all that’s left for me to add are some minor comments on spelling and syntax, that I will outline below.
Response: We do appreciate the comprehensive review that has clearly contributed to improve our paper over our original submission. The paper has been revised giving careful consideration to the points raised by the reviewer. We describe in detail below the changes made in each point.
Point 1: Introduction
Line 31. ‘…temperate climates, since this allows them to survive the unfavourable conditions of winter.’
Line 32. ‘…the meristems neither show growth nor development…’
Line 51. ‘…on the other hand, the statistical…’
Line 53. ‘…process behind dormancy has prevented an accurate delineation…’
Line 54. ‘…efforts have been made…’
Line 55. ‘…dormancy release during flower development...’? I’m not entirely sure what the authors are trying to say in this sentence.
Response 1: All sentences have been corrected as suggested (lines 31, 32, 51, 53-55).
Point 2: Results
Line 86. Though it is elaborated upon in the material and methods, for clarity I’d like to see a brief explanation of the ‘chilling portion’ when it is first mentioned.
Line 115. More principally, can one say that the ‘cultivars and seasons were the main causes of the variation’ when you don’t grow the specimens on both locations? I think one should grow all cultivars on both geographical locations to say whether location has an effect.
Line 140. There’s a sentence to be removed, from ‘this is a table…’ onwards.
Response 2:
- “Chill portions” has been described at first mention (Line 99).
- The sentence has been rewritten to avoid confusion (Lines 128-129).
- Extra text has been deleted in the Table 2 caption (Line 154).
Point 3: Discussion
Line 177. ‘…completely isolated from each other…’
Line 180. ‘…that continue their development to mature pollen…’
Line 183. ‘…resumed during ecodormancy, where meiosis indicates the transition…’
Line 239. ‘…difficult to obtain…’ as alternative to ‘…difficult to be available…’
Response 3:
All sentences have been corrected as suggested [lines 65, 68-69, 77-78 (this paragraph has been moved to Introduction following the suggestions of Reviewer 1), 251].
Point 4: In general, please be consistent in the use of ‘chill’ versus ‘chilling’ requirements (ignore this if I’m wrong, but I have the impression that they both are used for the same definition). I think you use ‘chilling requirements’ more regularly but I’ve come across some ‘chill’ too.
Response 4:
“Chill requirements” has been changed to “chilling requirements” throughout the manuscript.
Point 5: The discussion starts by the statement that ‘this approach provides a more objective and direct methodology to establish the agroclimatic requirements of apricot cultivars than previous methodologies’. While I agree with the statement, I think that it would be helped by the results that are now presented in lines 203-219. Is there a way to present the comparison in the ‘Results’ section? Paragraph 203-219 presents new data, which I believe is well suited for the ‘Results’ and it is further confirmation that the experiments/methodology is sound.
Response 5: Although the data of meiosis dates for each cultivar are included in figures 3 and 4 in “Results”, the specific data corresponding to ‘Orange Red’ have also been included in the text (lines 132-133).
